# Genetic Analysis and Sonography Characteristics in Fetus with SHOX Haploinsufficiency

**DOI:** 10.3390/genes14010140

**Published:** 2023-01-04

**Authors:** Lushan Li, Fang Fu, Ru Li, Xiangyi Jing, Qiuxia Yu, Hang Zhou, You Wang, Xin Yang, Min Pan, Jin Han, Li Zhen, Dongzhi Li, Can Liao

**Affiliations:** 1Prenatal Diagnostic Center, Guangzhou Women and Children’s Medical Center, Guangzhou Medical University, Guangzhou 510620, China; 2Eugenic and Perinatal Institute, Guangzhou Women and Children’s Medical Center, Guangzhou Medical University, Guangzhou 510620, China

**Keywords:** SHOX, CMA, WES, prenatal, short long bones, Xp22.33 microdeletion

## Abstract

Objective: SHOX haploinsufficiency have been commonly found in isolated short stature (ISS) and Léri–Weill dyschondrosteosis (LWD) patients. However, few publications have described the genetic analysis and clinical characteristics of fetuses with SHOX haploinsufficiency. Methods: Chromosomal microarray (CMA) were applied in 14,051 fetuses and sequentially whole exome sequence (WES) in 1340 fetuses who underwent prenatal diagnosis during 2016–2021. The analysis and summary of molecular genetics, sonographic characteristics, and follow-up results were performed in fetuses with SHOX haploinsufficiency without other genetic etiologies. A comparison was made between three groups according to prenatal diagnostic indications. Results: 8 (0.06%) fetuses of SHOX haploinsufficiency were all detected by CMA, of which 5 (62.5%) were detected with short long bones by ultrasound scan, and 4 were inherited from their previously undiagnosed parents. No pathogenic SHOX variants were found by WES. The detection rate of SHOX haploinsufficiency was obviously higher in the short long bone group (2.6%, 5/191) than the other abnormality group (0.03%, 1/3919) or no ultrasound abnormality group (0.02%, 2/9941). Three of the fetuses were liveborn with normal growth up to the age of four and four were terminated. Conclusion: The phenotype of fetuses with SHOX haploinsufficiency is highly varied. Over 1/3 of the cases exhibited no phenotype and nearly 2/3 with short long bones, in the absence of Madelung deformity during fetal development. SHOX haploinsufficiency should be considered in all antenatal presentations, especially in the case of isolated short long bones. CMA can provide effective detection.

## 1. Introduction

The SHOX gene is located at the tip of the short arms of both sex chromosomes, inside the telomeric portion of pseudoautosomal region 1 (PAR1), which contains genes that escape X inactivation. The SHOX gene encodes a transcription factor involved in the skeletal growth, and the function is dose-dependent, in that a loss of function mutation of one SHOX allele (haploinsufficiency) results in a SHOX deficiency, which then causes growth failure. SHOX deficiency contributes to the skeletal features in Turner syndrome. Despite its high penetrance, the highly variable clinical expression even within the same family [1], including Léri–Weill dyschondrosteosis (LWD), isolated short stature (ISS) and normal height, makes determining the prevalence of SHOX haploinsufficiency in the population extremely difficult. Given the results of studies of *SHOX* pathogenic variants in children with ISS and given that not all individuals with a *SHOX* pathogenic variant have short stature, it has been estimated that the prevalence of SHOX deficiency is at least 1:1000. Morbidity varies among different groups of people. Deletions and mutations of SHOX have been reported in 56–100% of patients with LWD and in 1.5–17% of children with idiopathic short stature (ISS) [1,2,3]. While there are still no data on prenatal cases, only very few fetuses related to SHOX haploinsufficiency have been reported [4,5].

The chromosomal microarrays (CMA) and whole exome sequencing (WES) are widely used to detect chromosomal abnormalities and monogenic genetic diseases in fetuses, adding to an increasing number of genetic diseases related to fetal phenotypes or incidentally detected prenatally. In this study, all fetal cases that underwent invasive prenatal diagnosis for CMA or CMA + WES in our center during the past six years were summarized. The detection rate, ultrasound characteristics and genetic mechanism of fetal cases with SHOX haploinsufficiency were analyzed as a reference for prenatal consultation, prognosis evaluation and appropriate postpartum follow-up.

## 2. Method

### 2.1. Study Cohort

All fetuses who underwent invasive prenatal diagnosis in Guangzhou Women and Children Medical Center and underwent CMA or sequentially WES during January 2016 to December 2021 were retrospectively analyzed. Cases with SHOX haploinsufficiency involving pathogenic copy number variations (CNVs) or pathogenic variations, such as single nucleotide variants (SNV), insertion and deletion (indel), were enrolled after excluding other genetic etiology. Fetal cases were divided into three groups according to the indication of invasive prenatal diagnosis: the group with short long bones, the group with other ultrasound abnormalities (including choroid plexus cysts, persistent right umbilical vein and other ultrasound soft indicators) and the group without any ultrasound structural abnormalities (the puncture indication was advanced maternal age, high risk of Down’s screening, or familial genetic factors, etc.). Genetic results and phenotype data were obtained by reviewing medical system records, following up via telephone, visiting patients on site and taking clinical photos. Literature-based reference ranges were used for fetal growth diameter and limb bones [6,7].

### 2.2. CMA

Informed consent was obtained from the pregnant women before the invasive procedure. CMA was performed by using an Affymetrix CytoScan HD/750K array with a single-nucleotide polymorphism array (SNP array) and array-based comparative genomic hybridization (aCGH) platforms at resolutions of 10 and 100 kb, respectively, according to the manufacturer’s protocol (Affymetrix Inc., Santa Clara, CA, USA). The built reference genome was aligned on GRCh37/hg19. Fetal DNA was extracted from chorionic villas, amniocytes, umbilical blood, or other family members’ peripheral lymphocytes by using a QiagenDNA Blood Midi/Mini Kit (Qiagen GmbH, Hilden, Germany). Invasive samples were analyzed with quantitative fluorescent polymerase chain reaction (QF-PCR) by utilizing a multiplex ligation-dependent probe amplification (MLPA) kit to exclude chromosomes 13, 18, 21, X, and Y and maternal cell contamination. Samples were subsequently subjected to CMA only when there was a normal QF-PCR result. The classification of CNVs was according to joint consensus recommendations of the American College of Medical Genetics and Genomics and ClinGen (Kearney et al., 2011; Riggs et al., 2020). The description of genomic findings identified by CMA was referred by the International System for Human Cytogenomic Nomenclature (ISCN 2020) (McGowan-Jordan Jean, 2020). The pathogenic CNVs, likely pathogenic CNVs, and variants of unknown significance (VOUS), were recorded and documented, but likely benign and benign VOUS were not considered. If a clinically significant variation or VOUS was identified in samples from the invasive procedure, parental CMA was recommended for these couples.

### 2.3. WES

Samples from fetus–parental trios with normal CMA results were subjected to ES. Exonic sequences were enriched in the DNA sample using Agilent SureSelect human exome capture probes (V6, Life Technologies, Guangzhou, China) according to the manufacturer’s protocol. The DNA library was sequenced on a HiSeq XTen or Illumina Novaseq 6000 system (Illumina, Inc., Guangzhou, China) to obtain 150 bp paired-end reads. Coverage for the samples was >99% at a 20× depth threshold. Additional details on the data analysis are provided in our previous study [8]. All the selected variants were then classified as pathogenic (P), likely pathogenic (LP), variant of unknown significance (VUS), likely benign, or benign according to the American College of Medical Genetics and Genomics (ACMG) guidelines [9]. All the qualifying variants were reviewed and classified by a multidisciplinary team that was comprised of clinical and molecular geneticists, genetic counselors, maternal fetal medicine physicians, neonatologists, bioinformaticians, and imaging experts to determine the relevance to the clinical presentation.

### 2.4. Statistic

Statistical analysis was performed by using IBM statistical program SPSS 25.0. The Chi-square test was used for categorical data. A *p*-value of less than 0.05 was considered statistically significant.

## 3. Result

### 3.1. Detection of SHOX Haploinsufficiency in Prenatal Diagnosis of Fetus

From 2016 to 2021, 14,051 fetal cases underwent invasive prenatal diagnosis according to varied indication and accepted CMA in our medical center, of which 191 fetuses (1.4%, 191/14,051) had short long bones, 9941 (70.7%) had other ultrasonic abnormalities excluding short long bones, and 3919 (27.9%) without structural abnormalities by ultrasound. A total of 332 cases with pathogenic CNVs, including the deletion of the SHOX gene, were detected, all of which were heterozygous deletions. There were 8 fetuses (0.06%, 8/14,051) with heterozygous deletion of Xp22.33 in CNVs with fragments ranging in size from 195 Kb to 2.2 Mb, which contained only the haploinsufficiency-sensitive pathogenic gene SHOX after excluding 317 cases of Turner syndrome and 7 cases combined with other genetic etiologies (Table 1). Among them, 4 cases were inherited from a previously undiagnosed parent, 2 cases were de novo, and 2 cases were unknown because their parents refused further testing. A total of 5 cases (62.5%, 5/8) had the phenotype of short long bones of limbs (Table 2). As a result of negative CMA results, 1340 cases were further tested by WES, and no pathogenic variants of the SHOX gene including SNPs and Indels were identified.

### 3.2. Difference in Detection Rate of SHOX Haploinsufficiency in Fetus with Varied Prenatal Diagnosis Indications

Fetal cases were divided into three groups according to varied prenatal diagnosis indications. In total, 5 cases (2.6%, 5/191) of SHOX haploinsufficiency were detected in the short long bone group. The detection rate was significantly higher than that in other ultrasound abnormalities groups (0.02%, 2/9941) and in the no-ultrasound-abnormality group (0.03%, 1/3919), statistically significant (the continuous correction X^2^ were 147.47 and 67.116 respectively, *p* < 0.001). There was no statistical difference between the no ultrasound abnormal group and other ultrasound abnormal groups (*p* > 0.05).

### 3.3. Follow-Up Results of Fetuses with SHOX Haploinsufficiency

Of the 8 cases with SHOX haploinsufficiency, 6 fetuses from 4 families were terminated due to the prenatal diagnosis results and ultrasound phenotypes, including 2 twins. One case was lost to follow-up. The remaining 3 cases inherited from their previously undiagnosed parents were born alive at term with a favorable outcome (Table 2).

Case 1 was male with a 440 kb deletion in Xp22.33 detected by CMA, which was inherited from his mother. The mother’s height is 153 cm with the typical LWD phenotype. It can be seen that the disproportionately short of the middle limb, and bilateral ulnar bones are protruded, forming Madelung’s deformity (see Figure 1). The mobility of both her wrists was normal, and she complained of no pain. The fetal experienced normal growth and development without skeletal malformation during pregnancy. He was delivered at 39 + 5 weeks of gestation, with normal height and weight. By the end of the follow-up, he was 6 years old with a height of 68 cm (−0.17 SD), a weight of 8.0 kg (−0.45 SD), and adequate motor development.

Case 5 was female with a 467 kb deletion in Xp22.33 detected by CMA, which was inherited from her father. Fetal ultrasound during pregnancy suggested that BPD and FL were less than −2 SD. Other long bones of the limbs (including tibia, fibula, humerus, ulna and radius) were around or slightly less than the 3rd percentile. The father is 165 cm tall with a family history of short stature. The paternal grandmother of case 5 measured 148 cm tall, but no genetic test was conducted. Neither of them had abnormal appearances, including limbs. The fetus was delivered at 37 + 6 weeks of gestation, with a birth length of 49 cm and a weight of 2.7 kg. His growth and development were similar to those of his age group until the age of four. There is no obvious disproportion or bending deformity in the limbs and abnormality in movement and intelligence.

Case 8 was male, and the CNV deletion region was xp22.33 del 532,467–1,216,144, which was inherited from his mother. The mother’s height is 146 cm. It can be seen that the long bones in the limbs are obviously short and the bilateral wrists are slightly malformed. The wrist mobility is normal. During pregnancy, the ultrasound showed that the fetus had short limbs and long bones (all around 3rd), and was delivered at 40 + 2 weeks of gestation, with a length of 46 cm (−2.59 SD) and a weight of 2.8 kg (−1.33 SD). He was followed up to 4 months of age, with a height of 61.5 cm, a weight of 6.25 kg, and normal motor development.

## 4. Discussion

The SHOX gene is located within the pseudoautosomal region of the X (Xp22.33) and Y (Yp11.3) chromosomes. SHOX-deficiency disorders present with a variable clinical phenotype of which the most consistent feature is short stature [10]. They are caused by SHOX haploinsufficiency inherited in a pseudoautosomal-dominant manner and follow the rules of autosomal inheritance. Despite SHOX haploinsufficiency’s high penetrance and failure of linear growth at an early age, it is rare to be diagnosed before late childhood, especially when there is no family history of the condition. It is clear from published reports that many cases of SHOX haploinsufficiency with a mild phenotype remain undiagnosed. Additionally, different inclusion criteria, cohort size, and genetic tests available for detecting pathogenic variants of SHOX and its enhancer regions may affect the detection of SHOX haploinsufficiency. In this study, we summarized all fetal cases who underwent invasive prenatal diagnosis for CMA and CMA + WES in our medical center, and it is the first time to review and analyze the occurrence of SHOX haploinsufficiency in population with varied prenatal diagnosis indications, as well as its ultrasound characteristics and molecular genetic results.

The detection rate of SHOX haploinsufficiency among 14,051 fetuses by CMA was 0.06%, and no LWD caused by SHOX homozygous deletion was detected. Surprisingly, in our study, no pathogenic variant was detected in the 1340 fetus by WES. A growing number of studies have shown that WES is a powerful supplement to traditional genetic testing for prenatal diagnosis, especially in cases of congenital skeletal malformations. The additional detection rate by WES can be 15.4–24% [4,5], and even up to 81–89% in several studies [11,12]. These findings suggest that the effectiveness of WES is high in fetuses with skeletal dysplasia. However, the majority of SHOX pathogenic variants are large deletions encompassing the entire gene; nearly 90% of cases were detected by CMA and only about 10% pathogenic variants were detected by sequence analysis [13], which may, to some extent, explain why all cases of SHOX haploinsufficiency were detected by CMA but not a single case by WES in our study. In addition, sample selection bias from the retrospective study may also have an impact. Generally, further WES is more likely to be accepted by couples with more severe fetal phenotypes. In this study, SHOX haploinsufficiency has only a mild prenatal phenotype of short long bones (mostly not less than or around—3rd) and even no ultrasound findings in some fetuses, which was consistent with previously reported cases [4,5]. Moreover, the capture chip of WES used in our study does not cover the enhancer regions, so the cases related to pathogenic variations of SHOX enhancer cannot be detected.

In our study, 37.5% of the SHOX haploinsufficiency cases had no structural malformations, while the remaining 62.5% had no other ultrasonic deformities but short long bones. These results are consistent with the currently reported cases, indicating that short long bones of the limbs are probably the only prenatal ultrasound findings in association with SHOX haploinsufficiency. It can be explained by the spatiotemporally restricted expression pattern of SHOX in human embryos. When the various bones of the arm can be identified (at Carnegie Stage 21), SHOX expression is mainly restricted to the middle portion of the arm, around the distal ends of the humerus, radius and ulna, and in a few bones of the wrist. Analogously, the results of SHOX in situ hybridization on the lower limbs resemble the expression pattern described for the upper limb development [13]. Obviously, the intrauterine phenotypic spectrum is simpler than that reported by postpartum patients from LWD at the severe end of the spectrum to a nonspecific short stature at the mild end of the spectrum. LWD typically develops in mid-to-late childhood with a four-fold higher prevalence in females than in males [14]. Higher estrogen levels have been proposed as a mechanism for the more severe symptoms in girls vs. boys [15]. It has been suggested that gonadal estrogens exert a maturational effect on skeletal tissues that are susceptible to unbalanced premature growth plate fusion and skeletal maturation due to SHOX haploinsufficiency, facilitating the development of skeletal lesions in a female-dominant and pubertal tempo-influenced fashion. In this study, there was no significant association between gender and intrauterine phenotype probably because during the fetal period, estrogen levels are much lower than during adolescence in both genders [16].

Isolated short long bones are commonly observed as a manifestation of intrauterine growth retardation (IUGR) [17]. It is worth noting that IUGR and SHOX haploinsufficiency can be present in the same case. In case 4, one twin displayed early-onset severe IUGR combined with SHOX haploinsufficiency. The difference from common IUGR is that the degree of short femur (−4.29 SD) is significantly higher than that of the biparietal diameter, head circumference and abdominal circumference (−2.75 SD~−3.75 SD), and the degree of the short long bones of limbs (all far less than 3rd) is more serious than that of another fetus in DCDA and another fetus with SHOX haploinsufficiency (3rd~50th–90th), suggesting that in IUGR fetuses with disproportionately short long bones of the limbs, the possibility of SHOX haploinsufficiency should be considered.

This study showed that half (4/8) of SHOX haploinsufficiency cases were inherited from previously undiagnosed parents with clinical features that were suggestive of SHOX deficiency. The height of affected parents ranged from −1.23 SD to −2.17 SD. Our study found no correlation of the embryonic phenotypes between the liveborn children with de novo SHOX haploinsufficiency and the liveborn children inherited from their parents. A noteworthy finding of our study was the fact that all the three liveborn cases with a favorable prognosis were inherited from their parents. It suggests that the genetic testing results may have an impact on the parents’ decisions on pregnancy since they are more likely to tolerate a fetus with a SHOX deficiency within their own genetic context after receiving adequate genetic counseling. In each of our inherited cases, there seems to be a correlation between the fetal long bone phenotype and the height of both parents when excluding combined IUGR. The parents of case 8 had a more severe short stature, and correspondingly, their fetus had more severe short long bones of the limbs. Conversely, both parents of case 1 were of a slightly short stature, and their fetus had normal growth and skeletal development during pregnancy and after birth. Further systematic studies were required.

Although no correlation has been established between the severity of the phenotypes and the underlying pathogenic variants based on our analysis and published literature [18,19,20,21], making it impossible to accurately predict the severity of phenotype based on prenatal molecular genetic testing results and prenatal counseling is still a challenge, there are currently effective treatments. For prepubertal children with SHOX-deficient short stature, recombinant human growth hormone (rhGH therapy) treatment can make a gain in final height of 7 to 10 cm [22,23]. For individuals with LWD and painful bilateral Madelung deformity, different operative procedures have been attempted to decrease pain and restore wrist function [24]. SHOX deficiency detected by prenatal diagnosis regardless of any indication facilitated the regular monitoring of postnatal growth and development and optimal timing of any growth hormone treatment.

## 5. Conclusions

Our research reported for the first time the detection rate of SHOX haploinsufficiency in a fetus by genetic techniques commonly used for prenatal diagnosis. Our experience suggests that over 1/3 of the cases with SHOX haploinsufficiency exhibited no prenatal phenotype and nearly 2/3 with short long bones, in the absence of Madelung deformity during fetal development. Given the incomplete penetrance and variable phenotype of SHOX deficiency, it should be carefully considered in all antenatal presentations, especially isolated short long bones. Parents of affected cases should be offered corresponding genetic testing. Whether their results and phenotypes prove to be an associated factor of intrauterine phenotypes will require more systematic studies. Nevertheless, there are currently viable treatments. No matter if the phenotype of SHOX deficiency appears to be intrauterine, timely monitoring and treatments can be carried out postnatally, and importantly, our research can provide reference for prenatal counseling.

## Figures and Tables

**Figure 1 genes-14-00140-f001:**
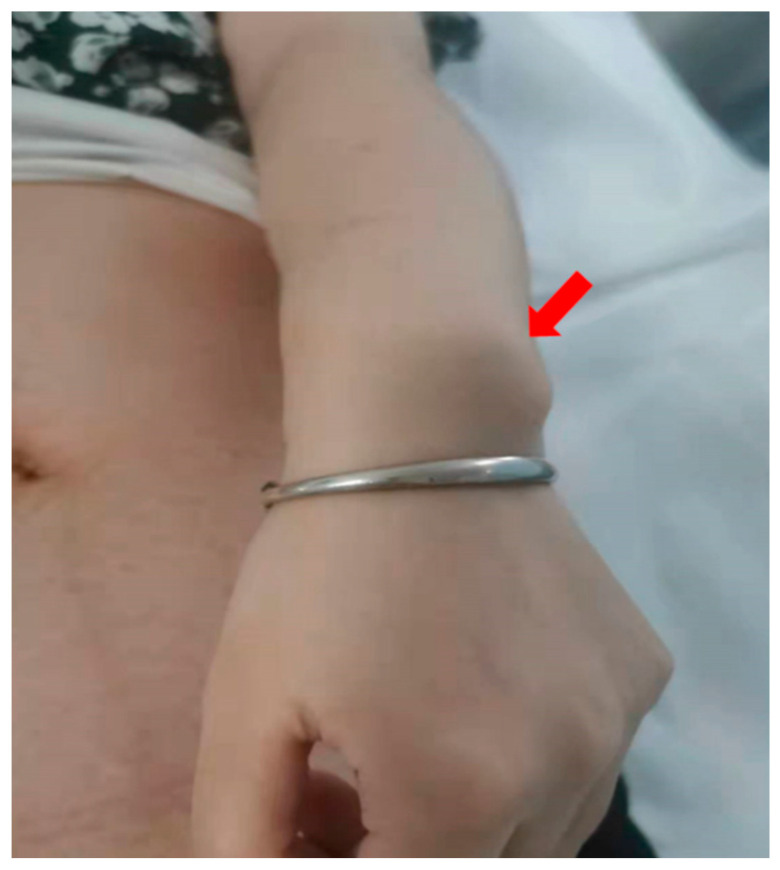
Madelung deformity of the wrist in the mother of case 1. Ulnar bones are protruded, forming Madelung deformity.

**Table 1 genes-14-00140-t001:** Summary of SHOX haploinsufficiency cases. Parent measurements of height are given in cm with corresponding standard deviation in brackets. F—female, M—male, MD—Madelung deformity, Mat—maternally inherited, Pat—paternally inherited, DN—de novo, UN—unknown.

Case	Gender	CNV	CNV SIZE	SHOX Enhancer	Origin	Mother Height	Father Height	Prenatal Diagnosis Indications	Pregnancy Outcome
1	M	Xp22.33 451,090–891,455 × 1	440 kb	contained	Mat	153 cm (−1.4 SD) With MD	170 cm (−0.45 SD)	high risk of T21 based on the first trimester serum screen	liveborn
2	M/F	Fetus A: Xp22.33 168,546–2,368,105 × 1 Fetus B:45, X	2.2 Mb	contained	UN	153 cm (−1.4 SD)	165 cm (−1.27 SD)	MCDA-Fetus A: persistent right umbilical vein Fetus B: Cystic hygroma, Bilateral pleural effusion, ascites	Termination
3	F	Xp22.33 168,546–1,597,685 × 1	1.43 Mb	contained	DN	160 cm (−0.1 SD)	170 cm (−0.45 SD)	NIPT:Sex chromosome abnormality	Termination
4	M/M	Xp22.33 del 480,573–756,010 × 1	275 kb	contained	Pat	158 cm (−0.48 SD)	160 cm (−2.10 SD)	DCDA-Fetal A: FGR Fetal B: normal	Termination
5	F	Xp22.33 del 372,012–839,488 × 1	467 kb	contained	Pat	155 cm (−1.4 SD)	165 cm (−1.27 SD)	Both BPD and FL < −2 SD	Liveborn
6	F	Xp22.33 del 484,176–679,369 × 1	195 kb	upstream enhancers contained	UN	158 cm (−0.48 SD)	172 cm (−0.12 SD)	Advanced Maternal Age, Choroid plexus cyst	Loss of follow-up
7	M	Xp22.33 del 387,397–679,369 × 1	291 kb	upstream enhancers contained	DN	166 cm (1.01 SD)	168 cm (−0.78 SD)	Short long bones (<−2 SD)	Termination
8	M	Xp22.33 del 532,467–1,216,144 × 1	684 kb	contained	Mat	146 cm (−2.73 SD) With MD	163 cm (−1.6 SD)	FL<−2 SD	Liveborn

**Table 2 genes-14-00140-t002:** Intrauterine and postnatal phenotypes of SHOX haploinsufficiency cases.

Case	Prenatal Ultrasound Findings	Gestation	Fetal Growth	Long Bones of Limbs	Postnatal Phenotype	Follow-Up
1	N	34 + 2	HC 32.9 (2.0 SD) BPD 9.3 (2.1 SD) AC 32.0 (1.78 SD) FL 6.3 (0.23 SD)	HL 6.01 (50th–90th) Rad 5.08 (50th–90th) Ulna 5.73 (50th–90th) Tib 5.91 (50th–90th) Fib 5.78 (50th–90th)	39 + 5 W H:52 cm (0.89 SD) W:4.0 Kg (1.66 SD)	6 month, H:68 cm (−0.17 SD) W:8.0 Kg (−0.45 SD) No phenotype up to follow-up
2	Fetus A of MCDA twins: persistent right umbilical vein	20 + 2	HC 17.1 (−0.23 SD) BPD 4.7 (0.04 SD) AC 15.1 (−0.01 SD) FL 2.9 (−0.95 SD)	-	Termination	-
	Fetal B: Cystic hygroma, Bilateral pleural effusion, ascites		HC 15.5 (−2.1 SD) BPD 4.3 (−1.51 SD) AC 21.7 (7.62 SD) FL 2.6 (−2.76 SD)	-		
3	Short long bones (<−2 SD)	24 + 3	HC 21.0 (−1.23 SD) BPD 5.8 (−0.84 SD) AC 18.9 (−0.68 SD) FL 3.5 (−2.97 SD)	HL 3.48 (<−3rd) Rad 2.84 (<−3rd) Ulna 3.35 (<−3rd) Tib 3.31 (<−3rd) Fib 3.15 (<−3rd)	Termination	-
4	Fetal A of DCDA twins: FGR	22 + 5	HC 17.1 (−2.75 SD) BPD 4.7 (−3.49 SD) AC 14.5 (−3.39 SD) FL 2.9 (−4.29 SD)	HL 2.8 (<−3rd) Rad 2.23 (<−3rd) Ulna 2.41 (<−3rd) Tib 2.61 (<−3rd) Fib 2.49 (<−3rd)	Termination	-
	Fetal B: N		HC 20.8 (0.60 SD) BPD 5.7 (0.73 SD) AC 18.8 (1.09 SD) FL3.7 (−0.32 SD)	HL 3.4 (10th) Rad 2.83 (10th) Ulna 3.35 (10th–50th) Tib 3.41 (10th–50th) Fib 3.29 (10th–50th)		
5	Both BPD and FL < −2 SD	37 + 4	HC 30.5 (−1.98 SD) BPD 8.0 (−3.17 SD) AC 32.8 (0.33 SD) FL 6.2 (−2.48 SD)	HL 5.58 (≈−3rd) Rad 4.56 (<−3rd) Ulna 5.31 (≈−3rd) Tib 5.61 (≈−3rd) Fib 5.56 (<−3rd)	37 + 6 W H: 49 cm (−0.41 SD) W: 2.7 Kg (−1.42 SD)	4 years H:101 cm (−0.54 SD) W:16.5 Kg (0.19 SD) No phenotype up to follow-up
6	Choroid plexus cyst	20 + 1	HC 16.9 (−0.18 SD) BPD 4.7 (0.09 SD) AC 14.8 (−0.10 SD) FL 3.1 (0.10 SD)	Unmeasured	Loss of follow-up	-
7	Short long bones (<−2 SD)	24 + 6	HC 22.26 (−0.44 SD) BPD 5.95 (−0.84 SD) AC 18.5 (−1.50 SD) FL 3.6 (−3.37 SD)	HL 3.43 (<−3rd) Rad 2.86 (<−3rd) Ulna 3.17 (<−3rd) Tib 3.28 (<−3rd) Fib 3.05 (<−3rd)	Termination	-
8	FL<−2 SD	33 + 3	HC 29.8 (−0.47 SD) BPD 8.2 (−0.63 SD) AC 27.7 (−0.69 SD) FL 5.42 (−2.81 SD)	HL 5.23 (−3rd–10th) Rad 4.14 (≈−3rd) Ulna 4.81 (≈−3rd) Tib 4.98 (<−3rd) Fib 4.82 (−3rd)	40 + 2 W H: 46 cm(−2.59 SD) W: 2.8 Kg (−1.33 SD)	4 months H: 61.5 cm (−1.34 SD) W: 6.25 Kg (−1.48 SD) No phenotype up to follow-up

Fetal measurements on antenatal scanning are given in cm with corresponding centiles or standard deviation in brackets.

## Data Availability

The original contributions presented in the study are included in the article, further inquiries can be directed to the corresponding author.

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
