# Peer review of "Genetic Analysis and Sonography Characteristics in Fetus with SHOX Haploinsufficiency"

_genes, 2023, doi:10.3390/genes14010140_

Round 1
Reviewer 1 Report
Dear Editor,
in this study, the authors wish to provide the mutational spectrum of SHOX gene in a large cohort of fetuses (n=14051). However, the results are presented in a confusing way and the tables are inapprehensible. Moreover, the introduction and the discussion sections misses to cite and discuss about the possible correlations that have been proposed in literature between the extention of the SHOX deletion (+/- including the enharcers and which enhancer) and the phenotype.
In my opinion, the presented cohort of cases has the potential of scientific value, but the results and discussion have to be otherwise presented. The tablles should also include which region of the gene is included in the rearrangment (enhancer/s??). Moreover, the prenatal diagnosis indication should be better clarified: for example, what does it mean "high risk of T21"? based on which test/parameter?
Also, the authors declare of not having found any pathogenic/likely pathogenic SHOX variant by WES. I believe that at least a list of the "hot VUS" should be provided and discussed in respect to the observed phenotype.
Author Response
Please see the attachment.
Point 1: the introduction and the discussion sections misses to cite and discuss about the possible correlations that have been proposed in literature between the extention of the SHOX deletion (+/- including the enharcers and which enhancer) and the phenotype.
Response 1: Thanks for your advice. Your suggestion is very good and hope to know more about the SHOX gene enhancer variation. Based on data from previous studies, the majority of SHOX pathogenic variants are large deletions encompassing the entire gene, that nearly 90% of cases were detected by CMA and only about 10% pathogenic variants were detected by sequence analysis. Of note, deletions with the SHOX enhancer regions that leave SHOX intact have been reported, with an unclear proportion. However, the capture chip of WES used in our study does not cover the above enhancer region, so the cases related to pathogenic variations of SHOX enhancer cannot be detected. The limitations of the above detection method have been supplemented in the text.
In addition, up to the end of our study, all CNVs regions of the 8 SHOX deletion cases contained the entire SHOX gene. There’s no CNVs duplication or deletion contained or overlapped only the SHOX enhancer. Therefore, I am so sorry that the correlation between variants/CNVs of SHOX enhancers and phenotypes cannot be analyzed in this study.
Recently, after the end of this study, there were two cases with Xp22.33 microduplication. These regions contained the last exon of SHOX gene and its downstream enhancer region, respectively genomic positions of them were chrX:602558-1234634 and chrX:614734-1356093. Neither of the two cases showed skeletal abnormalities, and were still unborn.
Identified in different studies, Genomic positions of SHOX enhancer elements SHOXa (NM 000451.3) are as follows: chrX:585,079 –607 558; chrX:516, 610 –517 229; chrX:460,279 – 460 664; chrX:398,357–398 906; chrX:714,085–714 740; chrX:750,825–751 850; chrX:780,580 –781 235; and chrX:834,746 – 835 548 (Tel;hg19,build37) [ doi:10.1210/er.2016-1036].
Point 2: The tablles should also include which region of the gene is included in the rearrangment (enhancer/s??).
Response 2: In our study, all 8 cases with SHOX gene deletion were detected by CMA. Their CNVs deletion regions contained the entire SHOX gene and it’s enhancer regions, except CNVs deletions in case 6 and case 7 which do not contain downstream enhancers of SHOX. I have revised and added this information in table 1 according to your suggestion.
Point 3: The prenatal diagnosis indication should be better clarified: for example, what does it mean "high risk of T21"? based on which test/parameter?
Response 3: Thanks for your advisement. I have supplemented more detailed prenatal diagnosis indications in the text, such as ‘high risk of T21 based on the first trimester serum screen’.
Point 4: Also, the authors declare of not having found any pathogenic/likely pathogenic SHOX variant by WES. I believe that at least a list of the "hot VUS" should be provided and discussed in respect to the observed phenotype.
Response 4: WES is a phenotype driven test. Among the 1340 cases underwent WES, 124 cases had skeletal abnormalities. I reviewed the results again and found no case of VUS related to SHOX gene variant phenotype. The following are all the fetal cases with variants of VUS detected by WES in this study.
The list of fetal cases with variants of uncertain significance (VUS) detected by WES.
Case ID |
Ultrasound finding |
Gene (OMIM ID) |
Transcript |
Nucleotide change |
Zygosity |
ACMG Classification |
Origin |
Disease (OMIM ID) |
Inheritance model |
P6149 |
Subependymal cysts,Dilation of lateral ventricles,Short long bone,ntrauterine growth retardation |
EVC (604831) |
NM_153717.2 |
c.1564-8C>T |
Het |
VUS |
De novo |
WEYERS ACROFACIAL DYSOSTOSIS(193530) |
AD |
P5866 |
Microcephaly |
CREBBP(600140) |
NM_004380.2 |
c.2393A>G |
Het |
VUS |
De novo |
RUBINSTEIN-TAYBI SYNDROME 1(180849) |
AD |
P5790 |
Dilation of lateral ventricles |
BRAF (164757) |
NM_004333.4 |
c.1061G>A |
Het |
VUS |
De novo |
CARDIOFACIOCUTANEOUS SYNDROME 1(115150) |
AD |
A21339 |
Agenesis of corpus callosum,Subependymal cysts |
ARX (300382) |
NM_139058.2 |
c.994C>G |
Het |
VUS |
Mat |
CORPUS CALLOSUM, AGENESIS OF, WITH ABNORMAL GENITALIA(300004)LISSENCEPHALY, X-LINKED, 2(300215) |
XLR |
P5652 |
Dilation of lateral ventricles |
FBN1 (134797) |
NM_000138.4 |
c.6379+4A>G |
Het |
VUS |
De novo |
MARFAN SYNDROME(154700) |
AD |
P4781 |
Dandy-Walker malformation |
ASXL1 (612990) |
NM_015338.5 |
c.1205G>A |
Het |
VUS |
De novo |
BOHRING-OPITZ SYNDROME(605039) |
AD |
P7481 |
Dilation of lateral ventricles |
TAF1 (313650) |
NM_001286074.1 |
c.1540C>T |
Het |
VUS |
Mat |
MENTAL RETARDATION, X-LINKED, SYNDROMIC 33(300966);DYSTONIA 3, TORSION, X-LINKED(314250) |
XLR |
QY33 |
Agenesis of corpus callosum |
L1CAM (308840) |
NM_000425.4 |
c.2254G>A |
Het |
VUS |
Mat |
MASA SYNDROME(303350) |
XLR |
P8246 |
Microcephaly |
TBCD (604649) |
NM_005993.4 |
c.1573C>T |
Het |
VUS |
Mat |
ENCEPHALOPATHY, PROGRESSIVE, EARLY-ONSET, WITH BRAIN ATROPHY AND THIN CORPUS CALLOSUM(617193) |
AR |
|
|
TBCD (604649) |
NM_005993.4 |
c.2178+5A>G |
Het |
VUS |
Pat |
|
|
NG306 |
Cerebellar vermis hypoplasia |
CEP120+ (613446) |
NM_153223.3 |
c.2132_2133delAA |
Het |
LP |
Mat |
JOUBERT SYNDROME 31(617761) |
AR |
|
|
CEP120 (613446) |
NM_153223.3 |
c.322-8T>C |
Het |
VUS |
Pat |
|
|
P5698 |
Dilation of lateral ventricles,dilation of the third ventricle,Subarachnoid hemorrhage |
BICD2(609797) |
NM_001003800.1 |
c.1991C>T |
Het |
VUS |
De novo |
SPINAL MUSCULAR ATROPHY, LOWER EXTREMITY-PREDOMINANT, 2B, PRENATAL ONSET(618291) |
AD |
A26839 |
Agenesis of corpus callosum |
SZT2 (615463) |
NM_015284.3 |
c.*926_*938delTCACCTCGAGGCT |
Hom |
VUS |
Mat |
EPILEPTIC ENCEPHALOPATHY, EARLY INFANTILE, 18; EIEE18(615463) |
AR |
|
|
SZT2 (615463) |
NM_015284.3 |
c.*2693_*2705delATCCCAGCATGGG |
Hom |
VUS |
Pat |
|
|
P5022 |
Cleft lip |
GLI2 (165230) |
NM_005270.4 |
c.880G>A |
Het |
VUS |
De novo |
CULLER-JONES SYNDROME(615849) |
AD |
P7050 |
Cleft palate,Cleft lip |
HSPG2 (142461) |
NM_001291860.1 |
c.6467G>A |
Het |
VUS |
Mat |
DYSSEGMENTAL DYSPLASIA, SILVERMAN-HANDMAKER TYPE(224410);SCHWARTZ-JAMPEL SYNDROME, TYPE 1(255800) |
AR |
|
|
HSPG2 (142461) |
NM_001291860.1 |
c.1189C>T |
Het |
VUS |
Pat |
|
|
A25868 |
Cleft lip |
TBX22 (300307) |
NM_001109878.1 |
c.1169C>A |
Het |
VUS |
Mat |
ABRUZZO-ERICKSON SYNDROME(302905) |
XL |
A26184 |
Cleft palate,Cleft lip |
FREM2 (608945) |
NM_207361.5 |
c.5237T>C |
Het |
VUS |
Mat |
FRASER SYNDROME 1(219000) |
AR |
|
|
FREM2 (608945) |
NM_207361.5 |
c.6322G>A |
Het |
VUS |
Pat |
|
|
NG56 |
Cleft palate |
LRP2 (600073) |
NM_004525.2 |
c.10027C>T |
Het |
VUS |
Mat |
DONNAI-BARROW SYNDROME(222448) |
AR |
|
|
LRP2 (600073) |
NM_004525.2 |
c.7306A>G |
Het |
VUS |
Pat |
|
|
A25001 |
Cleft palate,Cleft lip |
DHCR7 (602858) |
NM_001360.2 |
c.1060G>A |
Het |
VUS |
Pat |
SMITH-LEMLI-OPITZ SYNDROME(270400) |
AR |
|
|
DHCR7 (602858) |
NM_001360.2 |
c.289G>A |
Het |
VUS |
Mat |
|
|
A23804 |
Pulmonary sequestration,Congenital cystic adenomatoid malformation of the lung |
HSPG2 (142461) |
NM_001291860.1 |
c.11717G>T |
Het |
VUS |
Pat |
DYSSEGMENTAL DYSPLASIA, SILVERMAN-HANDMAKER TYPE(224410) |
AR |
|
|
HSPG2 (142461) |
NM_001291860.1 |
c.1078C>T |
Het |
VUS |
Mat |
|
|
A26335 |
Pulmonary sequestration |
STRA6 (610745) |
NM_001199042.1 |
c.436T>C |
Het |
VUS |
Pat |
MICROPHTHALMIA, SYNDROMIC 9(601186) |
AR |
|
|
STRA6 (610745) |
NM_001199042.1 |
c.367A>G |
Het |
VUS |
Mat |
|
|
A22460 |
Aplasia/Hypoplasia of the lungs,Hand polydactyly, Torticollis,Abnormality of the ribs,Abnormality of the thoracic spine,Mesocardia, Pulmonary artery sling,Atrial septal defect |
ZIC3 (300265) |
NM_003413.3 |
c.1306C>G |
Het |
VUS |
Mat |
HETEROTAXY, VISCERAL, 1, X-LINKED(306955) |
XLR |
A24830 |
Ventricular septal defect,Coarctation of aorta,Tricuspid regurgitation |
LONP1 (605490) |
NM_004793.3 |
c.2704-4C>A |
Het |
VUS |
Mat |
CODAS SYNDROME(600373) |
AR |
|
|
LONP1 (605490) |
NM_004793.3 |
c.686C>T |
Het |
VUS |
Pat |
|
|
A24266 |
Dextrocardia,Pulmonary artery atresia,Ventricular septal defect |
MMP21 (608416) |
NM_147191.1 |
c.557G>T |
Het |
VUS |
Mat |
HETEROTAXY, VISCERAL,7(608416) |
AR |
|
|
MMP21 (608416) |
NM_147191.1 |
c.551C>T |
Het |
VUS |
Pat |
|
|
P7846 |
Tetralogy of fallot |
FLNA (300017) |
NM_001110556.1 |
c.6002G>A |
Het |
VUS |
Mat |
CARDIAC VALVULAR DYSPLASIA, X-LINKED(314400) |
XL |
P6610 |
Single atrium,Complete atrioventricular canal defect |
MED12 (300188) |
NM_005120.2 |
c.1963A>G |
Het |
VUS |
Mat |
LUJAN-FRYNS SYNDROME(309520) |
XLR |
A25552 |
Transposition of the great arteries,Ventricular septal defect |
GPC3 (300037) |
NM_004484.3 |
c.338-6_338-5dupTT |
Het |
VUS |
De novo |
SIMPSON-GOLABI-BEHMEL SYNDROME, TYPE 1(312870) |
XLR |
A18455 |
Double outlet right ventricle,Ventricular septal defect,Coarctation of aorta |
NOTCH1(190198) |
NM_017617.4 |
c.4781G>A |
Het |
VUS |
De novo |
ADAMS-OLIVER SYNDROME 5(616028) |
AD |
P6063 |
Right aortic arch,Persistent left superior vena cava |
ABL1 (189980) |
NM_007313.2 |
c.2077C>T |
Het |
VUS |
De novo |
CONGENITAL HEART DEFECTS AND SKELETAL MALFORMATIONS SYNDROME(617602) |
AD |
NG323 |
Cardiomegaly,Tricuspid regurgitation, |
PRDM16(605557) |
NM_022114.3 |
c.1391C>T |
Het |
VUS |
De novo |
LEFT VENTRICULAR NONCOMPACTION 8(615373) |
AD |
A26870 |
Ventricular septal defect |
PKD1L1 (609721) |
NM_138295.3 |
c.6558+3A>G |
Het |
VUS |
Mat |
HETEROTAXY, VISCERAL, 8, AUTOSOMAL; HTX8() |
AR |
|
|
PKD1L1 (609721) |
NM_138295.3 |
c.3511G>A |
Het |
VUS |
Pat |
|
|
C4145 |
Omphalocele |
FREM1 (608944) |
NM_144966.5 |
c.5596C>G |
Het |
VUS |
Mat |
MANITOBA OCULOTRICHOANAL SYNDROME(248450) |
AR |
|
|
FREM1 (608944) |
NM_144966.5 |
c.4738C>T |
Het |
VUS |
Pat |
|
|
P8427 |
Peritoneal cystic mass |
PLVAP+ (607647) |
NM_031310.2 |
c.946C>T |
Het |
VUS |
Pat |
DIARRHEA 10, PROTEIN-LOSING ENTEROPATHY TYPE(618183) |
AR |
|
|
PLVAP (607647) |
NM_031310.2 |
c.298C>G |
Het |
VUS |
Mat |
|
|
P7258 |
Multicystic kidney dysplasia |
SNRPB (182282) |
NM_198216.1 |
c.455C>G |
Het |
VUS |
Mat |
CEREBROCOSTOMANDIBULAR SYNDROME(117650) |
AD |
P6183 |
Renal agenesis |
GATA3+ (131320) |
NM_001002295.1 |
c.59A>G |
Het |
VUS |
De novo |
HYPOPARATHYROIDISM, SENSORINEURAL DEAFNESS, AND RENAL DISEASE(146255) |
AD |
P5618 |
Renal duplication |
BMP4 (112262) |
NM_001347912.1 |
c.809G>A |
Het |
VUS |
De novo |
MICROPHTHALMIA, SYNDROMIC 6(607932) |
AD |
P7054 |
Multicystic kidney dysplasia |
NPHP3 (608002) |
NM_153240.4 |
c.2261C>A |
Het |
VUS |
Mat |
RENAL-HEPATIC-PANCREATIC DYSPLASIA 1(208540),]MECKEL SYNDROME, TYPE 7(267010) |
AR |
|
|
NPHP3 (608002) |
NM_153240.4 |
c.652A>C |
Het |
VUS |
Pat |
|
|
P6902 |
Renal agenesis |
DYNC2H1(603297) |
NM_001080463.1 |
c.9314G>A |
Het |
VUS |
Mat |
SHORT-RIB THORACIC DYSPLASIA 3 WITH OR WITHOUT POLYDACTYLY(613091) |
AR |
|
|
DYNC2H1(603297) |
NM_001080463.1 |
c.10063+8A>C |
Het |
VUS |
Pat |
|
|
P6757 |
Ectopic kidney |
IFT172 (607386) |
NM_015662.2 |
c.3925G>A |
Het |
VUS |
Pat |
SHORT-RIB THORACIC DYSPLASIA 10 WITH OR WITHOUT POLYDACTYLY(615630) |
AR |
|
|
IFT172(607386) |
NM_015662.2 |
c.2218C>T |
Het |
VUS |
Mat |
|
|
P8256 |
Hydronephrosis,Renal duplication,Micropenis |
LAS1L(300964) |
NM_031206.4 |
c.23G>A |
Het |
VUS |
Mat |
WILSON-TURNER X-LINKED MENTAL RETARDATION SYNDROME(309585) |
XLR |
NG365 |
Multicystic kidney dysplasia,Oligohydramnios |
PKHD1(606702) |
NM_138694.3 |
c.10058T>G |
Het |
VUS |
Pat |
POLYCYSTIC KIDNEY DISEASE 4 WITH OR WITHOUT POLYCYSTIC LIVER DISEASE(263200) |
AR |
|
|
PKHD1(606702) |
NM_138694.3 |
c.4798G>A |
Het |
VUS |
Mat |
|
|
P5898 |
Hydronephrosis,Renal dysplasia |
PIEZO2(613629) |
NM_022068.3 |
c.6437T>C |
Het |
VUS |
De novo |
MARDEN-WALKER SYNDROME(248700) |
AD |
P7278 |
Short long bone,Hand polydactyly |
ACAN(155760) |
NM_013227.3 |
c.703A>T |
Hom |
VUS |
Pat/Mat |
SPONDYLOEPIMETAPHYSEAL DYSPLASIA, AGGRECAN TYPE(612813) |
AR |
P6896 |
Talipes equinovarus,Abnormality of the wrist |
KDM5C(314690) |
NM_004187.3 |
c.856A>G |
Het |
VUS |
Mat |
MENTAL RETARDATION, X-LINKED, SYNDROMIC, CLAES-JENSEN TYPE(300534) |
XLR |
A22329 |
Talipes equinovarus |
CHRNG(100730) |
NM_005199.4 |
c.130G>A |
Het |
VUS |
Pat |
MULTIPLE PTERYGIUM SYNDROME, ESCOBAR VARIANT(265000) |
AR |
|
|
CHRNG(100730) |
NM_005199.4 |
c.1036-4_1036-3delCT |
Het |
VUS |
Mat |
|
|
P7484 |
Abnormality of the hand |
ATP7A(300011) |
NM_000052.6 |
c.4479G>C |
Het |
VUS |
Mat |
OCCIPITAL HORN SYNDROME(304150);SPINAL MUSCULAR ATROPHY, DISTAL, X-LINKED 3(300489) |
XLR |
A26934 |
Talipes equinovarus |
PLOD3(603066) |
NM_001084.4 |
c.1795A>T |
Het |
VUS |
Pat |
BONE FRAGILITY WITH CONTRACTURES, ARTERIAL RUPTURE, AND DEAFNESS(612394) |
AR |
|
|
PLOD3(603066) |
NM_001084.4 |
c.889C>G |
Het |
VUS |
Mat |
|
|
P7243 |
Intrauterine growth retardation |
CUL7(609577) |
NM_001168370.1 |
c.4115T>G |
Het |
VUS |
Mat |
THREE M SYNDROME 1(273750) |
AR |
|
|
CUL7(609577) |
NM_001168370.1 |
c.2855A>G |
Het |
VUS |
Pat |
|
|
P7789 |
Intrauterine growth retardation,Polyhydramnios |
PIGA(311770) |
NM_002641.3 |
c.1403A>G |
Het |
VUS |
Mat |
MULTIPLE CONGENITAL ANOMALIES-HYPOTONIA-SEIZURES SYNDROME 2(300818) |
XLR |
P7043 |
Intrauterine growth retardation |
CUL7(609577) |
NM_001168370.1 |
c.5071G>A |
Het |
VUS |
Pat |
THREE M SYNDROME 1(273750) |
AR |
|
|
CUL7(609577) |
NM_001168370.1 |
c.2234G>A |
Het |
VUS |
Mat |
|
|
P7496 |
Intrauterine growth retardation |
FLNB(603381) |
NM_001164317.1 |
c.3661A>G |
Het |
VUS |
Mat |
SPONDYLOCARPOTARSAL SYNOSTOSIS SYNDROME(272460) |
AR |
|
|
FLNB(603381) |
NM_001164317.1 |
c.7762G>A |
Het |
VUS |
Pat |
|
|
P8481 |
Intrauterine growth retardation |
DOCK6(614194) |
NM_020812.3 |
c.5227G>A |
Het |
VUS |
Pat |
ADAMS-OLIVER SYNDROME 2; AOS2(614219) |
AR |
|
|
DOCK6(614194) |
NM_020812.3 |
c.4579C>T |
Het |
VUS |
Mat |
|
|
P7949 |
Pleural effusion,Hydrops fetalis,Polyhydramnios,Absence of stomach bubble on fetal sonography |
RFWD3+(614151) |
NM_018124.3 |
c.879G>T |
Het |
VUS |
Mat |
FANCONI ANEMIA, COMPLEMENTATION GROUP W(617784) |
AR |
|
|
RFWD3(614151) |
NM_018124.3 |
c.835A>G |
Het |
VUS |
Mat |
|
|
P7661 |
Ascites,Pericardial effusion |
LAMB2+(150325) |
NM_002292.3 |
c.4304C>T |
Het |
VUS |
Pat |
PIERSON SYNDROME(609049) |
AR |
|
|
LAMB2(150325) |
NM_002292.3 |
c.3339G>T |
Het |
VUS |
Mat |
|
|
C4881 |
Increased nuchal translucency |
RYR1(180901) |
NM_000540.2 |
c.2203C>T |
Het |
VUS |
Pat |
MINICORE MYOPATHY WITH EXTERNAL OPHTHALMOPLEGIA(255320) |
AR |
|
|
RYR1(180901) |
NM_000540.2 |
c.4610C>A |
Het |
VUS |
Mat |
|
|
A19240 |
Increased nuchal translucency |
PIEZO1(611184) |
NM_001142864.3 |
c.7130-8C>T |
Hom |
VUS |
De novo |
LYMPHATIC MALFORMATION 6(616843) |
AR |
C5357 |
Cystic hygroma |
LZTR1(600574) |
NM_006767.3 |
c.593+2T>C |
Het |
LP |
Pat |
NOONAN SYNDROME 2(605275) |
AR |
|
|
LZTR1(600574) |
NM_006767.3 |
c.1785+1G>A |
Het |
VUS |
Mat |
|
|
C5137 |
Increased nuchal translucency |
CEP290(610142) |
NM_025114.3 |
c.6806T>C |
Het |
VUS |
Mat |
MECKEL SYNDROME, TYPE 4(611134) |
AR |
|
|
CEP290(610142) |
NM_025114.3 |
c.4962_4963delAA |
Het |
LP |
Pat |
|
|
C5469 |
Cystic hygroma |
IGF1R(147370) |
NM_000875.4 |
c.43G>A |
Het |
VUS |
Pat |
INSULIN-LIKE GROWTH FACTOR I, RESISTANCE TO; IGF1RES(270450) |
AD/AR |
|
|
IGF1R(147370) |
NM_000875.4 |
c.3723-4G>A |
Het |
VUS |
Mat |
|
|
A26580 |
Increased nuchal translucency |
CC2D2A(612013) |
NM_001080522.2 |
c.1484G>A |
Het |
VUS |
Mat |
COACH SYNDROME(216360), MECKEL SYNDROME, TYPE 6; MKS6(612284), JOUBERT SYNDROME 9; JBTS9(612285) |
AR |
|
|
CC2D2A(612013) |
NM_001080522.2 |
c.4186A>G |
Het |
VUS |
Pat |
|
|
C5435 |
Increased nuchal translucency |
RPL10(312173) |
NM_001256577.2 |
c.-176G>T |
Het |
VUS |
Mat |
MENTAL RETARDATION, X-LINKED, SYNDROMIC, 35; MRXS35(300998) |
XLR |
P5932 |
Ventriculomegaly,Hypoplasia of fetal nasal bone,Short middle phalanx of the 5th finger |
TP63(603273) |
NM_003722.4 |
c.1318C>G |
Het |
VUS |
De novo |
ECTRODACTYLY, ECTODERMAL DYSPLASIA, AND CLEFT LIP/PALATE SYNDROME 3(604292)ï¼›RAPP-HODGKIN SYNDROME(129400) |
AD |
P5905 |
Renal duplication, Hydronephrosis |
FLNA(300017) |
NM_001110556.1 |
c.13C>T |
Het |
VUS |
De novo |
OTOPALATODIGITAL SYNDROME, TYPE II(304120);FRONTOMETAPHYSEAL DYSPLASIA 1(305620);MELNICK-NEEDLES SYNDROME(309350) |
XLR,XLD |
P4660 |
Cystic hygroma,Abnormality of the skeletal system,Polyhydramnios |
FLNB(603381) |
NM_001164317.1 |
c.2323+5G>C |
Het |
VUS |
De novo |
ATELOSTEOGENESIS, TYPE I(108720);ATELOSTEOGENESIS, TYPE (108721);LARSEN SYNDROME(150250) |
AD |
P4317 |
Dilation of lateral ventricles,Talipes equinovarus,Abnormality of finger |
BRPF1(602410) |
NM_001003694.1 |
c.2864G>A |
Het |
VUS |
De novo |
INTELLECTUAL DEVELOPMENTAL DISORDER WITH DYSMORPHIC FACIES AND PTOSIS(617333) |
AD |
A23916 |
Pulmonic stenosis, Enlargement of posterior fossa |
ZIC3(300265) |
NM_003413.3 |
c.159_161dupCGC |
Het |
VUS |
Mat |
HETEROTAXY, VISCERAL, 1, X-LINKED(306955) |
XLR |
A23793 |
Bilateral cleft lip and palate,Micrognathia,Single ventricle,Pulmonic stenosis,Situs inversus totalis |
FLNA(300017) |
NM_001110556.1 |
c.560A>G |
Het |
VUS |
Mat |
CARDIAC VALVULAR DYSPLASIA, X-LINKED(314400) |
XLR |
A23206 |
Pyelectasia,Talipes equinovarus |
AP4B1(607245) |
NM_006594.4 |
c.1591C>T |
Het |
VUS |
Pat |
SPASTIC PARAPLEGIA 47(614066) |
AR |
|
|
AP4B1(607245) |
NM_006594.4 |
c.1136C>T |
Het |
VUS |
Mat |
|
|
P6642 |
Holoprosencephaly,Talipes equinovarus,Cleft palate,Cleft lip |
ABCD1(300371) |
NM_000033.3 |
c.1367G>C |
Het |
VUS |
Mat |
ADRENOLEUKODYSTROPHY(300100) |
XLR |
A22165 |
Hypoplasia of fetal nasal bone,Left aortic arch with retroesophageal right subclavian artery,Hyperechogenic liver,Microphallus |
FGD1(300546) |
NM_004463.2 |
c.1967G>T |
Het |
VUS |
Mat |
AARSKOG-SCOTT SYNDROME(305400) |
XLR |
P5819 |
Left atrial isomerism,Double outlet right ventricle,Ventricular septal defect |
DYNC2H1(603297) |
NM_001080463.1 |
c.6044G>A |
Het |
VUS |
Mat |
SHORT-RIB THORACIC DYSPLASIA 3 WITH OR WITHOUT POLYDACTYLY(613091) |
AR |
|
|
DYNC2H1(603297) |
NM_001080463.1 |
c.6857C>T |
Het |
VUS |
Pat |
|
|
P8215 |
Absent radius,Hypospadias, Persistent left superior vena cava,Single umbilical artery |
CCDC22(300859) |
NM_014008.4 |
c.1840C>T |
Het |
VUS |
Mat |
RITSCHER-SCHINZEL SYNDROME 2(300963) |
XLR |
P8210 |
Thickened skin,Abnormal posturing,Open mouth |
ABCA12(607800) |
NM_173076.2 |
c.6858delT |
Het |
LP |
Pat |
ICHTHYOSIS, CONGENITAL, AUTOSOMAL RECESSIVE 4B(242500) |
AR |
|
|
ABCA12(607800) |
NM_173076.2 |
c.3977-4C>T |
Het |
VUS |
Mat |
|
|
P8373 |
Dilation of lateral ventricles,Pleural effusion,Complete atrioventricular canal defect,Double outlet right ventricle,Coarctation in the transverse aortic arch,Hypoplastic left heart |
PIEZO1(611184) |
NM_001142864.3 |
c.5214+7C>T |
Het |
VUS |
Mat |
LYMPHATIC MALFORMATION 6(616843) |
AR |
|
|
PIEZO1(611184) |
NM_001142864.3 |
c.3590T>C |
Het |
VUS |
Pat |
|
|
P8512 |
Ventricular septal defect, Hyperechogenic kidneys |
KIF14(611279) |
NM_014875.2 |
c.1672A>G |
Het |
VUS |
Pat |
MECKEL SYNDROME 12; MKS12(616258), MICROCEPHALY 20, PRIMARY, AUTOSOMAL RECESSIVE; MCPH20(617914) |
AR |
|
|
KIF14(611279) |
NM_014875.2 |
c.1408T>A |
Het |
VUS |
Mat |
|
|
P8470 |
Scoliosis, Tethered cord, Spina bifida, Hemivertebrae, Renal agenesis, Mesocardia, Single umbilical artery |
SLC25A22(609302) |
NM_001191061.1 |
c.-172_-170dupGCG |
Hom |
VUS |
Pat&Mat |
EPILEPTIC ENCEPHALOPATHY, EARLY INFANTILE, 3; EIEE3(609304) |
AR |
A26676 |
Transposition of the great arteries,Abnormality of the pinna |
MED13L(608771) |
NM_015335.4 |
c.5456C>T |
Het |
VUS |
Mat |
TRANSPOSITION OF THE GREAT ARTERIES(608808),MENTAL RETARDATION AND DISTINCTIVE FACIAL FEATURES WITH OR WITHOUT CARDIAC DEFECTS(616789) |
AD |

Reviewer 2 Report
Overall interesting study that correlates SHOX CMA findings with prenatal phenotypes (or lack thereof).
1. In Abstract Results, include that no pathogenic SHOX variants were found by whole exome sequencing.
2. Please provide a little more information on your methods for searching medical records for SHOX results. Were they organized so that you could just search for CNVs including SHOX? Or did you have to manually review every pathogenic result?
3. Did your method detect VUS that overlap SHOX gene? We see a sizable number of duplications that overlap the SHOX gene in postnatal patients. I suspect most of these are benign, but it would be of interest to readers to know how prevalent this finding is and if any of the patients have a phenotype that could suggest pathogenicity (gene interrupted by duplication).
4. Line 160. You introduce the abbreviation "LMD" but I do not see that it was previously defined? Did you intend this to mean Langer mesomelic dysplasia or LWD? Please clarify.
5. Line 166 and 183. I would include height standard deviation in parentheses (or make reference to the Table where it can be found).
6. Line 201. "LMD again." Clarify meaning.
7. Line 278 "exhibit no phenotype" - I would change to "exhibit no prenatal phenotype" for clarity.
Round 2
Reviewer 1 Report
I suggest of upload the table entitled "The list of fetal cases with variants of uncertain significance (VUS) detected by WES." as supplementary material.